# Processing Alaska Pollock Protein (*Theragra chalcogramma*) into Kamaboko Protein Mitigates Elevated Serum Cholesterol and Postprandial Glucose Levels in Obese Zucker *fa/fa* Rats

**DOI:** 10.3390/foods11213434

**Published:** 2022-10-29

**Authors:** Natsuka Takada, Ryota Hosomi, Kenji Fukunaga

**Affiliations:** 1Technical Research Department, Ichimasa Kamaboko Co., Ltd., 7-77, Tsushimaya, Higashi-ward, Niigata City 950-8735, Niigata, Japan; 2Faculty of Chemistry, Materials, and Bioengineering, Kansai University, 3-3-35, Yamate-cho, Suita City 564-8680, Osaka, Japan

**Keywords:** Alaska pollock, kamaboko, cholesterol, glucose tolerance, Zucker *fa/fa* rat

## Abstract

Fish paste products such as kamaboko (KB) are traditional Japanese foods prepared from fish meat. The health-promoting properties of fish proteins and their lysates include improving glucose and lipid metabolism. Although the KB manufacturing process and quality have been examined, limited studies have reported the health-promoting function of KB. This study aimed to evaluate the effects of processing Alaska pollock protein (APP) into KB protein (KBP) on serum lipids levels and postprandial glucose tolerance. Obese male Zucker *fa/fa* rats were fed on different diets for 4 weeks as follows: APP group, fed on a diet in which APP constituted 25% of total protein intake; KBP group, fed on a diet with APP-processed protein as the protein source; control group, fed on a diet with 100% casein as the protein source. Compared with those in the control group, the serum total cholesterol (TC) level was reduced and the elevated postprandial blood glucose level was mitigated during the high-carbohydrate meal tolerance test in the APP and KBP groups. Further, KBP exerted significantly higher effects on serum TC levels and glucose tolerance than APP.

## 1. Introduction

The consumption of marine products is reported to be epidemiologically and clinically beneficial for maintaining and improving health. Fatty fish contains high levels of functional nutrients, such as *n*-3 polyunsaturated fatty acids (PUFA), eicosapentaenoic acid (EPA), and docosahexaenoic acid (DHA) [1,2]. Epidemiological studies have demonstrated that the incidence of metabolic syndromes, such as obesity, hyperlipidemia, and type 2 diabetes mellitus (T2DM) is low among individuals and populations with high fish intake [3,4,5]. In particular, fish diets play an important role in the development of T2DM [6,7].

The beneficial effects of the consumption of fish and seafood can be attributed to the functions of *n*-3 PUFA, such as EPA and DHA. However, fish consumption does not coincide with the health benefits of refined fish oil [8,9,10,11,12,13]. Systematic reviews examining the correlation between fish consumption and the development of T2DM have reported conflicting results [14,15,16,17,18]. These results may be explained, to some extent, by differences in fish consumption or fish oil as a supplement, the place of residence of subjects and their eating habits, and the type and quantity of fish consumed.

Previously, we focused on proteins, a major component of fish as well as lipids, and demonstrated that Alaska pollock (AP) protein alters the composition of the gut microbiota and improves insulin sensitivity [19] and cholesterol-related metabolism in rats [20]. The health-promoting effects of fish proteins [19,20,21,22,23,24,25,26,27,28,29,30], baked fish [31,32], hydrolysates, peptides produced by digestion [33,34,35,36,37], water-soluble protein [38], and fish residues [39] have been examined in humans and experimental animals. However, the suppression of elevated postprandial blood glucose levels or related events by processed fish paste products, such as kamaboko (KB) has not been previously reported. Additionally, limited studies have examined other health functions of KB [40,41]. This study examined the effects of processing fish into KB, a traditional Japanese food, on serum lipid levels and blood glucose levels. AP, which is used worldwide as processed food, such as surimi for fish paste products, was selected as the protein source in this experiment. This study aimed to investigate the beneficial effects of processing AP protein (APP) into KB protein (KBP) on human health, especially on serum lipid concentration and glucose tolerance, and to publicize the health benefits of KB, a traditional Japanese food.

## 2. Materials and Methods

### 2.1. Experimental Diet

Obese Zucker *fa/fa* rats are less efficient at protein utilization, especially during the growth phase, which adversely affects their protein sufficiency. Therefore, Zucker *fa/fa* rats must consume more proteins to maintain adequate protein requirements during growth [42]. In this study, the American Institute of Nutrition (AIN)-93G diet [43] with 20% protein was used instead of the AIN-93M diet with 15% protein. The control diet comprised 20% casein in the AIN-93G formula, whereas the experimental diets comprised 15% casein and 5% APP or KBP. Table 1 shows the composition of experimental diets based on the AIN-93G formula. The diets differed only in their protein source.

To prepare APP, AP fillets whose skin was removed were cut into sticks, frozen in liquid carbon dioxide, freeze-dried, finely chopped, and powdered in a grinder. The crude fat in AP powder was removed with *n*-hexane/ethanol (2:1; *v/v*), and the residue was vacuum-dried and stored at −35 °C. The resulting powder was named APP.

To prepare KBP, KB was first prepared from raw AP fillets using the traditional method. AP meat was homogenized in a cooking blender at low temperatures and rinsed with copious amounts of cold water to remove soluble proteins. Three % (*w/w*) NaCl was added to the remaining AP meat and kneaded into a paste. Subsequently, the paste was molded into a plate shape, and steamed at 95 °C for 20 min. The prepared KB was cut into sticks after cooling, frozen in liquid carbon dioxide, freeze-dried, finely chopped, and powdered using a grinder. As KB has high salt content, it was desalted with cold water and freeze-dried. Additionally, the freeze-dried samples were defatted and vacuum-dried using the same methods as those used for processing APP and stored at −35 °C. The resulting powder was considered KBP.

A carbohydrate-rich diet for the meal tolerance test (MTT) was prepared as a modified semi-purified diet based on the AIN-93G formula with 400 g of sucrose/kg diet (composition of the diet for MTT: 225 g of casein, 70 g of soybean oil, 400 g of sucrose, 193 g of corn starch, 50 g of cellulose, 35 g of the mineral mix, 10 g of vitamin mix, 3 g of L-cysteine, 1.5 g of L-methionine, 2.5 g of choline bitartrate, and 10 g of maintenance supplement containing vitamin B_12_ and vitamin K_1_) as recommended by Reeves [44]. The experimental diet components were purchased from Oriental Yeast Co., Ltd. (Tokyo, Japan) and Fujifilm Wako Pure Chemical Co., Ltd. (Osaka, Japan).

### 2.2. Nutritional Component Analysis of Experimental Diets

Nutritional component analysis of the protein sources (casein, APP, and KBP) was performed following the routine methods. Briefly, water content was determined by heating the sample at 105 °C for 2 h under atmospheric pressure, while the crude ash content was determined by heating the sample at 550 °C for 24 h. Crude fat content was quantified with continuous extraction using a Soxhlet extractor. The micro Kjeldahl method was used to quantify crude protein content (nitrogen to protein conversion factor = 6.25) [45]. The amino acid composition was determined by hydrolyzing the protein in 6 M HCl at 110 °C for 24 h, followed by derivatization with phenylthiohydantoin [46]. Derivatized sample was loaded into a high-performance liquid chromatography (HPLC, Shimadzu Co., Kyoto, Japan) system on Shim-pack ODS column (3 µm, 250 × 3.0 mm i.d.; Shimadzu Co.) for detection equipped with an SPD-20A UV-Vis detector set to 269 nm wavelength [46]. For EPA and DHA determination, total lipids were extracted using the Bligh and Dyer method [47] and hydrolyzed to free fatty acids. The EPA and DHA contents of total lipids were derivatized with 9-anthryldiazomethane [48]. Derivatized sample was loaded into a HPLC (Shimadzu Co.) system on Shim-pack XR C8 (2.2 µm, 100 × 3.0 mm i.d.; Shimadzu Co.) for detection equipped with a RF-20A fluorescence detector set to 365 nm excitation and 412 nm emission wavelengths. For protein profiling, sodium dodecyl sulfate-polyacrylamide gel electrophoresis (SDS-PAGE) was performed using 12.5% polyacrylamide gels and a molecular weight marker (Bio-Rad Laboratories Inc., Hercules, CA, USA) as described previously [49]. Proteins in the gels were stained with Coomassie Brilliant Blue R-250 (Fujifilm Wako Pure Chemical Co., Ltd.).

### 2.3. Ethics Approval

The Animal Ethics Committee approved the protocol for experimental animals of Kansai University (Approval No. 2112) by the “Guide for the Care and Use of Experimental Animals” issued by the Prime Minister’s Office of Japan.

### 2.4. Experimental Animal and Care

Zucker *fa/fa* rats, which were used as experimental animals, exhibit aberrant lipid metabolism (a common pathological manifestation of obesity in humans) and are the most commonly used representative rat model for studying metabolic complications and potential treatments for obesity [7]. Male Zucker *fa/fa* rats aged 8 weeks were purchased from CLEA Japan Inc. (Tokyo, Japan). The animals were allowed to acclimatize for 7 days. Next, the rats were divided into three groups (8 rats) with comparable mean bodyweight (BW). The experimental period started when the rats achieved a BW of 350 g (standard error [SE] < 7 g, *n* = 8). The rats were weighed weekly during the experimental period. The housing conditions for rats, which were housed in cages, were as follows: circadian cycle, 12-h light-dark cycle (light from 08:00 to 20:00); constant temperature, 20–22 °C; relative humidity, 65 ± 15%. All rats had free access to tap water and experimental diets (ad libitum). A new diet and water were provided, and food and water intake and BW were measured every other day.

### 2.5. MTT

The MTT was performed using a carbohydrate-rich diet after the animals were allowed to fast overnight (12 h) on day 22 of the intervention. The rats were housed individually under fasting conditions from 20:00 to 08:00 h and allowed to feed a diet corresponding to 2 g sucrose/kg BW. After measuring the BW before the diet load, the dosage was calculated. The rats were allowed a maximum of 15 min to consume the meal. All rats consumed the meal within 15 min. After fasting and at 30, 60, 90, 120, 150, and 180 min post-carbohydrate-rich diet feeding, the dorsal tail vein was punctured to collect the blood samples in a plastic tube without an anticoagulant. The samples were centrifuged at 1500× *g* for 15 min to obtain the serum. The serum glucose and insulin levels were measured using the Glucose CII Test Wako (Fujifilm Wako Pure Chemical Co. Ltd., Osaka, Japan) and insulin enzyme-linked immunosorbent assay kit (Morinaga Institute of Biological Science, Inc., Tokyo, Japan), respectively.

### 2.6. Dissection

At the end of the feeding period (28 days), rats were euthanized. The blood samples were collected from the abdominal aorta under anesthesia (2–4% isoflurane; Fujifilm Wako Pure Chemicals Co., Ltd.) mixed with air. Before sampling, the animals were allowed to fast for 12 h and had free access to tap water. The serum was obtained by centrifuging the blood samples at 1500× *g* for 15 min. The liver, heart, kidneys, spleen, perirenal white adipose tissue (WAT), epididymal WAT, and thigh skeletal muscle were dissected and weighed. Tissues were frozen in liquid nitrogen and stored at −70 °C until analysis. Fecal samples were collected daily from days 23 to 28 of the intervention period. All samples were frozen in liquid nitrogen and stored at −70 °C until analysis.

### 2.7. Analysis of Biochemical Parameters in the Serum, Liver, and Feces

The serum aspartate aminotransferase (AST), alanine aminotransferase (ALT), triglyceride (TG), total cholesterol (TC), high-density lipoprotein-cholesterol (HDL-C), low-density lipoprotein-cholesterol (LDL-C), non-esterified fatty acids (NEFA), and total bile acid (TBA) levels were measured by a commercial organization (Oriental Yeast Co. Ltd. Tokyo, Japan).

Total liver lipids were extracted using the Bligh and Dyer method [47] and dissolved in isopropanol. The TG and cholesterol levels were determined using the colorimetric method with Triglyceride E Test Wako and Cholesterol E Test Wako (Fujifilm Wako Pure Chemical Co. Ltd.), respectively.

To extract fecal lipids, the fecal samples were extracted with nine volumes of 99.5% ethanol at 70 °C for 1 h. The supernatant was collected after centrifugation at 3500× *g* for 15 min. The centrifugation step was repeated three times. The pooled supernatant was aliquoted and used as a sample. The TBA and cholesterol contents in the sample were measured using the Total bile acid Test Wako and Cholesterol E Test Wako (Fujifilm Wako Pure Chemical Co. Ltd.), respectively.

### 2.8. Statistical Analysis

Data are expressed as mean ± SE. The means were compared using analysis of variance, followed by Tukey’s multiple comparison test. The cut-off level for statistical significance was set at a probability of 0.05. Statistical tendency was also set at 0.05 ≤ *p* < 0.10.

## 3. Results

### 3.1. Nutritional Composition of Casein, APP, and KBP

Table 2 shows the nutritional composition of casein, APP, and KBP. Compared with casein, APP and KBP exhibited higher contents of alanine, arginine, aspartic acid (aspartic acid + asparagine), and glycine but lower contents of proline. Additionally, the lysine to arginine and methionine to glycine ratios in APP and KBP were lower than those in casein. Trace amounts of EPA and DHA were detected in APP (24 and 66 mg/100 g, respectively) and KBP (19 and 43 mg/100 g, respectively). Similarly, trace amounts of cholesterol were detected in casein (14.5 mg/100 g), APP (15.6 mg/100 g), and KBP (14.3 mg/100 g).

Figure 1 shows the protein band patterns of casein, APP, and KBP subjected to SDS-PAGE. The two prominent bands of casein at approximately 29 kDa represent α-casein and β-casein. Meanwhile, APP and KBP exhibited prominent bands at approximately 200 and 45 kDa, respectively, which are myofibrillar proteins containing myosin (heavy chain, approximately 200 kDa) and actin (approximately 45 kDa). Additionally, the band patterns of APP and KBP were similar, except for the lack of myosin light chain 3 in KBP.

### 3.2. Growth Parameters and Organ Weight

Table 3 shows the growth parameters and organ weights of animals in different groups. The rats in the control, APP, and KBP groups exhibited similar BW at the beginning of rearing (350 g [SE < 7 g, *n* = 8]). The mean BW at weeks 1, 2, and 3 (data not shown) or at the endpoint were not significantly different between the groups. Furthermore, diet and water intake and fecal output were not markedly different between the groups. Similarly, the relative weight of organs, including the liver, heart, kidneys, spleen, perirenal WAT, epididymal WAT, and thigh skeletal muscle, at dissection were not significantly different between the groups. 

### 3.3. Biochemical Parameters

Table 4 shows the biochemical parameters of the serum, liver, and feces. The AST and ALT levels (liver function indicators) were not significantly different between the groups. Compared with those in the control group, the serum TC and HDL-C levels were significantly lower in the KBP group and tended to be lower in the APP group. The LDL-C levels in the APP and KBP groups were slightly lower than those in the control group. However, the serum TG and TBA levels were not significantly different between the groups. The levels of NEFA, a risk factor for the development of insulin resistance, in the APP and KBP groups were lower than those in the control group. The liver TG and cholesterol concentrations were not significantly different between the groups. Furthermore, the fecal TBA and cholesterol excretion levels were not significantly different between the groups.

### 3.4. MTT

Figure 2 shows the serum glucose concentrations as a function of time during MTT in obese Zucker *fa/fa* rats fed on experimental diets. The fasting glucose concentrations were not significantly different between the groups before MTT, indicating that APP and KBP do not affect fasting glucose levels. After the start of MTT, the glucose concentration increased rapidly in all groups up to 30 min and peaked at 30 min. At 30 and 60 min, the glucose concentration in the KBP group was significantly lower than that in the control and APP groups. However, at 30 min, the glucose concentrations in all groups began to gradually decrease. The serum glucose concentration in the APP group at 30 min lower than that in the control group. From 30 min to 120 min, the serum glucose concentrations in the KBP group were lower than those in the control group. The glucose concentrations at 180 min were similar to those at the start of MTT. Additionally, the area under the curve of serum glucose concentration in the KBP group was significantly lower than that in the control and APP groups (Figure 2B).

Figure 3 shows the serum insulin concentrations as a function of time during MTT in obese Zucker *fa/fa* rats fed on experimental diets. After the start of MTT, the insulin concentrations in all groups linearly increased for up to 60 min and peaked at 60 min. After 60 min, the insulin concentrations in all groups began to gradually decrease. The insulin concentrations at 180 min were similar to those at the start of MTT. Additionally, the insulin concentrations during MTT were not markedly different between the groups. Furthermore, the AUC of serum insulin concentrations was not significantly different between the groups during MTT.

## 4. Discussion

Most animal studies examining the effect of fish protein intake on glucose tolerance or lipid metabolism have often used fish protein as the sole protein source [22,24,25,50]. However, limited studies have examined the effects of a combination of fish protein and casein in the diet [13,26,38]. The feeding of fish protein as the sole dietary protein source may not meet the essential amino acid requirements of growing rats because the contents of branched-chain amino acids, leucine, isoleucine, valine, and phenylalanine in fish protein are lower than those in casein [23,24]. However, this study demonstrated that the amino acid composition was not significantly different between casein, APP, and KBP, indicating that processing AP into KB did not affect amino acid composition (Table 2). Furthermore, as the ratio of APP to KBP in the experimental diet was 25% (*w/w*), the amino acid composition was not markedly different between the control and experimental diets. Additionally, diet and water intake and BW gain were similar between the groups. Therefore, similar to casein, APP and KBP are utilized for growth and have high protein efficiency even though the proteins used in this study were not evaluated for digestibility.

In this study, obese Zucker *fa/fa* rats fed on a diet containing 25% APP or KBP derived and/or processed from AP fillets exhibited reduced serum TC and HDL-C levels. This is consistent with the results of our previous study, which reported the reduction in liver cholesterol and serum TC concentrations in Wistar rats fed on APP [20,21]. Fish oil is reported to reduce the serum TC levels in Zucker *fa/fa* rats via the activity of *n*-3 PUFA [51]. However, trace amounts of EPA and DHA were detected in the experimental proteins. Similarly, trace amounts of cholesterol content were detected in experimental proteins with no marked difference between groups, indicating that these components do not affect cholesterol metabolism. The serum cholesterol-lowering effect can be explained by several mechanisms. In particular, the serum cholesterol-lowering effect is mediated by promoting the excretion of sterols in the feces and altering cholesterol metabolism in the liver [52]. In this study, the serum TBA levels and fecal excretion of cholesterol and TBA were not significantly different, suggesting that the reduction in serum levels of TC and HDL-C in the APP and KBP groups did not result from increased fecal excretion of cholesterol and TBA. These results are consistent with those of a previous study [37], which reported that feeding diets containing hydrolyzed salmon protein as the sole protein source to Zucker *fa/fa* rats reduced the serum levels of TC and HDL-C. Additionally, the fecal cholesterol and TBA excretion in the two groups was not significantly different from that in the control group fed on a diet containing casein as the sole protein source. Endogenous cholesterol synthesis downregulation is one of the beneficial effects of dietary amino acids. Previous studies have reported that cholesterol synthesis is induced by the reductions in lysine/arginine and methionine/glycine ratios of the constituent amino acids of dietary proteins [37,53]. However, the lysine/arginine and methionine/glycine ratios were not significantly different between the diets. The cholesterol-lowering effect in the KBP group was higher than that in the APP group. Although this study did not examine the effects of cooked fish meat on cholesterol metabolism, Vikøren et al. [31] reported that the serum and tissue *n*-3 fatty acid content and composition were not markedly different in Zucker *fa/fa* rats fed on grilled and raw salmon fillets. These results indicated that heating fish meat did not affect the functional properties. Therefore, we believe that the stronger cholesterol-lowering effect of KBP relative to APP may be due to the functional properties imparted by processing AP meat into KB. However, the mechanism of the cholesterol-lowering effect of APP or KBP intake is unknown. Hence, there is a need to measure the expression levels of cholesterol metabolism-related genes in the liver.

Diet plays an important role in the development of T2DM [6,7]. Fish diets have been reported to improve glucose tolerance and lipid metabolism. However, systematic reviews examining the correlation between fish consumption and the development of T2DM have often reported conflicting results [14,15,16,17,18]. These results can be explained, to some extent, by differences in fish species, refined supplements, and the place of residence of subjects and their eating habits. One aim of this study was to determine the effect of processing AP meat into KB on the functions related to postprandial glucose tolerance. Glucose tolerance tests are of the following two types: the oral glucose tolerance test (OGTT), glucose is administered alone; MTT, a sample containing all nutrients is administered. The nutritional intake during MTT is more balanced than that during OGTT. Therefore, MTT accurately detects the effects of functional components in models of insulin resistance [54,55]. As humans spend most of their time after eating, observing the course of glucose regulation after MTT is more interesting than observing fasting glucose concentration. The clinical usefulness of MTT in treating patients with T2DM has been previously demonstrated [56,57]. Therefore, this study conducted MTT in obese Zucker *fa/fa* rats, which is a widely used model of genetic obesity that spontaneously develops aberrations resembling human metabolic syndromes, such as dyslipidemia, insulin resistance, mild glucose intolerance, and hyperinsulinemia [58,59], to evaluate the functional characteristics related to glucose tolerance, which is altered by processing AP meat into KBP.

Fasting glucose concentrations were similar in the groups before MTT, indicating that APP and KBP did not influence fasting glucose regulation. However, the KBP diet promoted significantly smaller increases in blood glucose at 30 and 60 min than the control and APP diets. The APP diet non-significantly suppressed the upregulation of blood glucose when compared with the control diet (Figure 2B). Thus, processing APP into KBP further enhanced glucose tolerance. and improved postprandial glucose regulation. In contrast, blood insulin concentration increased up to 60 min in response to the rise in blood glucose levels, followed by a gradual decrease. The blood insulin concentration was similar between groups, indicating that APP and KBP did not affect insulin concentration. As insulin concentrations in obese Zucker *fa/fa* rats are high even after fasting, reduced uptake of blood glucose into the liver and skeletal muscle is unlikely to be due to defective insulin secretion. King et al. [60] reported that insulin resistance in the skeletal muscle of obese Zucker *fa/fa* rats is severe due to the failure of glucose transporter translocation. Tremblay et al. [22] and Pilon et al. [25] revealed that feeding cod protein to rats decreased fasting blood glucose and insulin concentrations and postprandial blood glucose levels and improved peripheral insulin sensitivity when compared with casein feeding. We believe that this is a contradiction observed in this study because this study used Zucker *fa/fa* rats, a pathological model of obesity. Nobile et al. [35] reported that cod protein hydrolysate improved human insulin sensitivity and promoted glucagon-like peptide-1 (GLP-1) and cholecystokinin (CKK) secretion. Additionally, Tremblay et al. [22] suggested that cod protein normalizes the insulin-mediated activation of the phosphatidylinositol 3-kinase/protein kinase B pathway and improves glucose transporter type 4 translocation in the skeletal muscle of high-fat diet-induced obesity rat model. Therefore, the suppression of hyperglycemia by KBP intake may have improved glucose uptake into the liver and skeletal muscle cells via gastrointestinal hormones, such as GLP-1 and CKK.

KB is prepared by cutting the muscle of white fish, such as AP, into small pieces, followed by washing in cold water, the addition of 2–3% salt to solubilize the salt-soluble proteins and allow the formation of a mesh structure, and heating the myofibrillar proteins to enable the formation of a gel that gives it unique elasticity. SDS-PAGE analysis revealed similar band patterns of APP and KBP, except for the lack of myosin light chain 3 in KBP, indicating that myofibrillar proteins, such as myosin and actin are the major components (Figure 1). Although SDS-PAGE analysis provides information on protein composition, it does not confirm the presence of actomyosin (the substance responsible for gel formation), which exhibits reticular structures. However, the main reason for improved glucose tolerance and other functional properties is the unique elasticity obtained by processing AP meat into KB. Recent studies have reported that microbiota has an important role in human health and diseases [61,62,63]. Additionally, different types of proteins in the diet differentially affect gut microbiota composition. Previously, we evaluated the correlation between insulin sensitivity, glucose tolerance, and gut microbiota after APP administration in vivo [19]. The results demonstrated that compared with casein, APP altered the structure of the fecal microbiota and markedly increased the relative abundance of the genera *Blautia* and *Akkermansia*. The elasticity of KBP is due to the braided structure of the gel, which provides different digestive properties and may improve gut microbiota composition. Therefore, the improvement in glucose tolerance observed in this study with KBP intake can be attributed to the improvement of gut microbiota composition.

## 5. Conclusions

This study aimed to determine the effect of processing AP meat into KB on serum lipids levels and postprandial glucose tolerance in obese Zucker *fa/fa* rats. The animals in the APP and KBP groups were fed on a diet in which APP and KBP constituted 25% of the total protein intake, respectively. Meanwhile, 100% casein served as the protein source for the animals in the control group. Compared with the control, APP and KBP exerted beneficial effects on serum cholesterol and postprandial blood glucose levels. Additionally, processing AP into KB improved glucose tolerance. Therefore, a diet containing 25% fish protein may be more practical and beneficial to humans than the previously reported diet in which fish protein constituted 100% or 50% of the protein source. Additionally, KB, a traditional Japanese food, can be potentially used as an additive in functional foods and dietary supplements for patients with hypercholesterolemia or blood glucose metabolism disorders. Therefore, future studies on fish protein intake must use small amounts of fish protein, preferably over an extended period.

## Figures and Tables

**Figure 1 foods-11-03434-f001:**
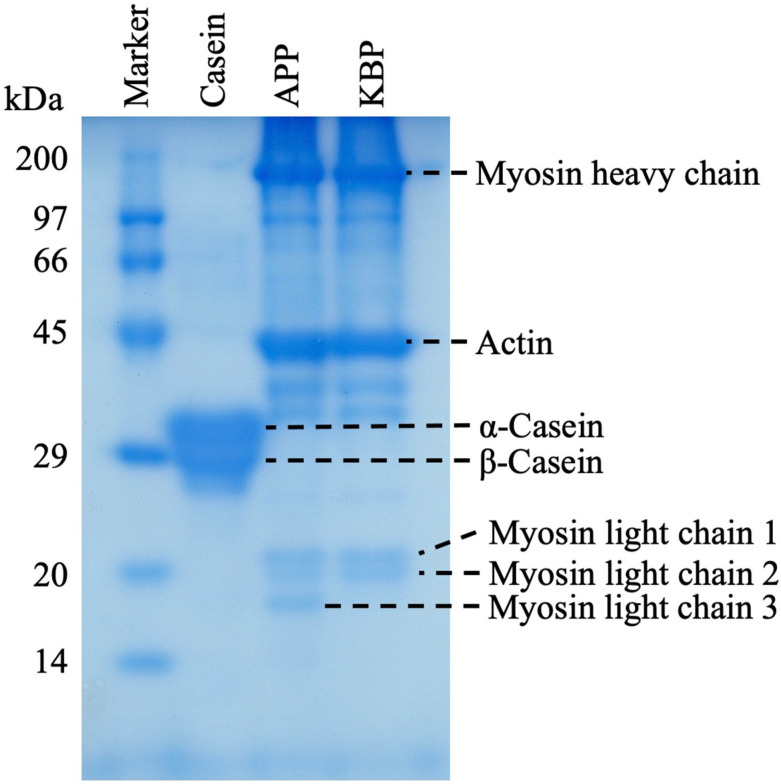
Sodium dodecyl sulfate-polyacrylamide gel electrophoresis (SDS-PAGE) analysis of casein, APP, and KBP band patterns. SDS-PAGE was performed using a 12.5% polyacrylamide separation gel. APP, Alaska pollock protein; KBP, kamaboko protein.

**Figure 2 foods-11-03434-f002:**
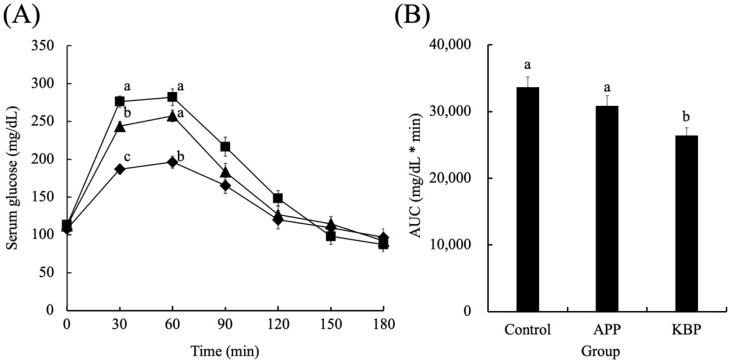
Serum glucose concentrations as a function of time (**A**) and area under the curve (**B**) during MTT. Control (■), APP (▲), and KBP (◆) groups. Data represent mean ± standard error from the data of 8 animals per group. Means were compared using analysis of variance, followed by Tukey’s multiple comparison test with a cut-off level of 0.05 probability of significance. Different letters of the same item indicate significant differences. APP, Alaska pollock protein; AUC, area under the curve; KBP, kamaboko protein; MTT, meal tolerance test.

**Figure 3 foods-11-03434-f003:**
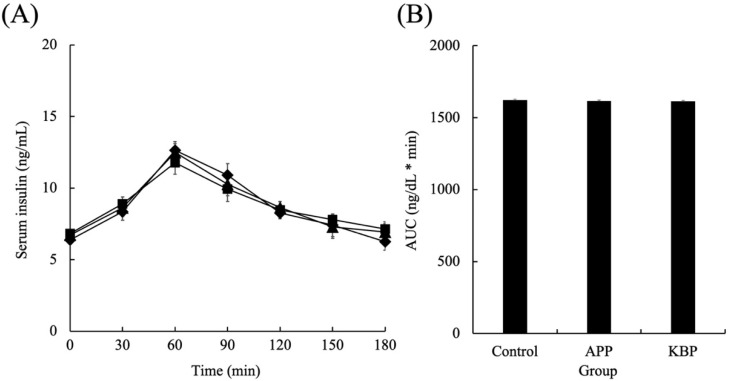
Serum insulin concentrations as a function of time (**A**) and area under the curve (**B**) during MTT. Control (■), APP (▲), and KBP (◆) groups. Data represent mean ± standard error from the data of 8 animals per group. Means were compared using analysis of variance, followed by Tukey’s multiple comparison test with a cut-off level of 0.05 probability of statistical significance. Different letters of the same item indicate significant differences. APP, Alaska pollock protein; AUC, area under the curve; KBP, kamaboko protein; MTT, meal tolerance test.

**Table 1 foods-11-03434-t001:** Composition of the experimental diets.

	Control ^1^	APP	KBP
	g/kg		
Casein	200	150	150
Alaska pollack protein		50	
Kamaboko protein			50
Dextrinized corn starch	132	132	132
Corn starch	397.486	397.486	397.486
Sucrose	100	100	100
Cellulose	50	50	50
Choline bitartrate	2.5	2.5	2.5
L-Cystine	3	3	3
AIN-93 vitamin mixture	10	10	10
AIN-93G mineral mixture	35	35	35
Soybean oil	70	70	70
*tert*-Butylhydroquinone	0.014	0.014	0.014

^1^ Diet was prepared based on the American Institute of Nutrition (AIN-93G) formula. APP, Alaska pollack protein; KBP, kamaboko protein.

**Table 2 foods-11-03434-t002:** Nutritional compositions of casein, APP, and KBP.

	Experimental Proteins		
	Casein	APP	KBP
Crude protein (g/100 g)	92.4	91.8	90.5
Amino acid compositions (wt%)			
Alanine	2.9	5.6	4.9
Arginine	4.5	7.4	7.5
Aspartic acid ^1^	6.4	10.1	9.4
Glutamic acid ^2^	22.2	18.3	19.2
Glycine	2.4	4.5	4.2
Histidine	3.0	2.6	2.3
Isoleucine	4.6	4.3	4.5
Leucine	10	7.7	8.7
Lysine	6.5	8.9	9.4
Methionine	2.3	3.5	3.2
Phenylalanine	4.8	4.3	4.6
Proline	9.5	3.5	3.6
Serine	5.5	4.5	4.2
Threonine	4.2	5.3	4.6
Tyrosine	4.8	3.5	3.5
Valine	6.5	6.0	6.1
Lysine/Arginine ratio	1.434	1.20	1.25
Methionine/Glycine ratio	0.96	0.78	0.76
Crude fat (g/100 g)	1.3	0.3	0.2
EPA (mg/100 g)	-	24	19
DHA (mg/100 g)	-	66	43
Cholesterol (mg/100 g)	14.5	15.1	13.7
Moisture (g/100 g)	2.6	3.1	2.4
Crude ash (g/100 g)	1.8	4.3	3.2

^1^ Aspartic acid: Aspartic acid + asparagine. ^2^ Glutamic acid: glutamic acid + glutamine. APP, Alaska pollock protein; DHA, docosahexaenoic acid; EPA, eicosapentaenoic acid; KBP, kamaboko protein.

**Table 3 foods-11-03434-t003:** Effect of dietary casein, APP, and KBP on growth parameters and organ weights in rats.

	Experimental Groups		
	Control	APP	KBP
Growth parameters			
Initial BW (g)	352.9 ± 4.3	352.7 ± 4.1	353.2 ± 4.1
Final BW (g)	486.3 ± 8.7	491.5 ± 9.2	492.0 ± 6.9
BW gain (g/day)	133.4 ± 2.4	138.8 ± 4.0	138.8 ± 2.7
Diet intake (g/kg BW/24 h)	73.5 ± 1.8	75.7 ± 2.1	74.9 ± 1.7
Water intake (g/kg BW/24 h)	58.4 ± 2.3	56.8 ± 2.5	58.1 ± 2.6
Fecal output (dry g kg BW/24 h)	5.2 ± 0.6	5.5 ± 0.8	5.3 ± 0.7
Organ weight			
Liver (g/kg BW)	34.7 ± 0.9	34.2 ± 0.7	35.1 ± 1.0
Heart (g/kg BW)	2.65 ± 0.08	2.67 ± 0.09	2.70 ± 0.08
Kidneys (g/kg BW)	5.29 ± 0.17	5.18 ± 0.14	5.23 ± 0.13
Spleen (g/kg BW)	1.63 ± 0.05	1.59 ± 0.04	1.60 ± 0.05
Perirenal WAT (g/kg BW)	36.3 ± 0.5	37.5 ± 0.6	36.8 ± 0.5
Epididymal WAT (g/kg BW)	32.6 ± 0.5	33.5 ± 0.4	32.1 ± 0.4
Thigh skeletal muscle (g/kg BW)	3.42 ± 0.12	3.53 ± 0.09	3.47 ± 0.11

Data represent mean ± standard error from the data of eight animals per group. APP, Alaska pollock protein; BW, body weight; KBP, kamaboko protein; WAT, white adipose tissue.

**Table 4 foods-11-03434-t004:** Effect of dietary casein, APP, and KBP on the rat serum, liver, and fecal biochemical parameters.

	Experimental Groups		
	Control	APP	KBP
Serum biochemical parameter			
AST (IU/L)	58.9 ± 1.5	60.7 ± 1.2	63.1 ± 1.7
ALT (IU/L)	36.8 ± 0.9	35.4 ± 0.8	36.0 ± 0.8
TG (mg/dL)	305.7 ± 36.8	274.4 ± 32.3	268.3 ± 26.1
TC (mg/dL)	197.1 ± 10.6 ^a^	168.2 ± 7.2 ^b^	146.5 ± 8.5 ^c^
HDL-C (mg/dL)	139.3 ± 6.8 ^a^	128.3 ± 6.4 ^a^	109.9 ± 3.7 ^b^
LDL-C (mg/dL)	28.2 ± 1.3	26.3 ± 1.1	25.1 ± 1.2
NEFA (mEq/L)	0.89 ± 0.03 ^a^	0.67 ± 0.04 ^b^	0.54 ± 0.03 ^c^
TBA (μmol/L)	19.1 ± 1.1	19.6 ± 1.3	20.3 ± 0.9
Liver lipid parameters			
TG (mg/g liver)	112.3 ± 8.5	108.6 ± 10.3	106.3 ± 9.4
Cholesterol (mg/g liver)	2.7 ± 0.5	2.5 ± 0.6	2.4 ± 0.5
Fecal lipid parameters			
Cholesterol (mg/24 h)	4.1 ± 0.3	3.9 ± 0.3	4.3 ± 0.4
TBA (μmol /24 h)	7.7 ± 0.6	7.3 ± 0.7	7.5 ± 0.6

Data represent mean ± standard error from the data of eight animals per group. The means were compared using analysis of variance, followed by Tukey’s multiple comparison test with a probability of statistical significance of 0.05. Different letters for the same item indicate significant differences. ALT, alanine aminotransferase; AST, aspartate aminotransferase; APP, Alaska pollock protein; HDL-C, high-density lipoprotein-cholesterol; KBP, kamaboko protein; LDL-C, low-density lipoprotein-cholesterol; NEFA, non-esterified fatty acids; TBA, total bile acid; TC, total cholesterol; TG, triglyceride.

## Data Availability

The corresponding author shall share the data presented in this study upon reasonable request.

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
