# Peer review of "Processing Alaska Pollock Protein (Theragra chalcogramma) into Kamaboko Protein Mitigates Elevated Serum Cholesterol and Postprandial Glucose Levels in Obese Zucker fa/fa Rats"

_foods, 2022, doi:10.3390/foods11213434_

Round 1
Reviewer 1 Report
The authors investigated the health-promoting properties of kamaboko proteins include improving glucose and lipid metabolism. The results suggested that 25% KBP intake is functionally sufficient to mitigate the elevated serum TC level and postprandial blood glucose levels. This work seemed to be interesting, however, more evidences are needed to reveal the underlying mechanisms.
(1) As the processing products of fish protein, no significant differences in compositions between Alaska pollock protein and kamaboko in essential. How to explain their distinct health-promoting properties?
(2) In methods section, the equipment information should be provided.
(3) The present evidences may be difficult to the obtain the conclusion “Thus, 25% KBP intake is functionally sufficient to mitigate the elevated serum TC level and 23 postprandial blood glucose levels.”, because the most data show no significant differences. More evidences should be provided to support it.
Author Response
Dear Sir,
Thank you very much for your letter with regard to our manuscript (foods-1972523) together with the comments.
We tried to revise the manuscript as much as possible in line with the suggestions reviewer. Our answers of the reviewers’ suggestions are as follows:
The authors investigated the health-promoting properties of kamaboko proteins include improving glucose and lipid metabolism. The results suggested that 25% KBP intake is functionally sufficient to mitigate the elevated serum TC level and postprandial blood glucose levels. This work seemed to be interesting, however, more evidences are needed to reveal the underlying mechanisms.
Thank you for your helpful suggestion. We will describe the items you have pointed out.
(1) As the processing products of fish protein, no significant differences in compositions between Alaska pollock protein and kamaboko in essential. How to explain their distinct health-promoting properties?
While it is clear from the experimental results that there is no significant difference between the actual composition of pollock protein and kamaboko, this paper focuses on the improvement of functional properties of pollock muscle when processed into kamaboko. It is a well-known fact that the physical properties of fish meat change significantly when processed into kamaboko. Proteins solubilized by salt form a three-dimensional network structure and become gelatinous with unique elasticity when heated. As a result, the digestive process in the gastrointestinal tract produces peptides and other digested products that function differently from mere fish meat and may have different effects on the organism and intestinal bacteria. We recognize that further research is needed. The mechanistic expression of the functionality is mentioned and discussed in the DISCUSSION section, citing what has been revealed in previous studies and papers published by other researchers.
(2) In methods section, the equipment information should be provided.
We have added (L105-L108 and L111- L 114) a description of the equipment, HPLC columns used in the analysis.
(3) The present evidences may be difficult to the obtain the conclusion “Thus, 25% KBP intake is functionally sufficient to mitigate the elevated serum TC level and 23 postprandial blood glucose levels.”, because the most data show no significant differences. More evidences should be provided to support it.
As you point out, most of the data do not show significant differences, but the relevant data do show that it is useful in alleviating elevated serum TC and postprandial blood glucose levels, as stated in the title of the paper 25% KBP intake is an overstatement "functionally sufficient" and has been deleted.
The English text has been proofread by a native English-speaking professional (Editage) for a fee (L454-L455).

Reviewer 2 Report
The title is confusing since the use of the word “protein” makes the reader infer that the study will focus on a specific protein isolated from the fish species (Alaska pollock); however, the word “protein” is used interchangeably as muscle tissue, and both concepts are considerably different. This reviewer considers that “muscle tissue” should be used throughout the manuscript when appropriate.
Please add the scientific name of the fish species in the title.
Line 12: What do the authors refer to with “gelatinized fish protein”? How is this phenomenon defined? Are proteins able to undergo gelatinization?
Table 3: Although it is explained in the text, it would be illustrative to display Tukey’s comparison results by adding superscript letters to the means. The same applies to all the parameters shown in Table 4 and Figures 2 and 3.
Author Response
Dear Sir,
Thank you very much for your letter with regard to our manuscript (foods-1972523) together with the comments.
We tried to revise the manuscript as much as possible in line with the suggestions reviewer. Our answers of the reviewers’ suggestions are as follows:
(1) The title is confusing since the use of the word “protein” makes the reader infer that the study will focus on a specific protein isolated from the fish species (Alaska pollock); however, the word “protein” is used interchangeably as muscle tissue, and both concepts are considerably different. This reviewer considers that “muscle tissue” should be used throughout the manuscript when appropriate.
Thank you for pointing this out. The term "fish protein" in this paper refers to the general view of fish meat as a food, not a specific protein. In our previous papers and those of other researchers, fish protein refers to the edible part of fish, i.e., muscle tissue, so we think it is fine as it is. We have corrected some parts (L12) where “meat” would be better than “protein”.
(2) Please add the scientific name of the fish species in the title.
We have included the scientific name alongside as you indicated (L2).
(3) Line 12: What do the authors refer to with “gelatinized fish protein”? How is this phenomenon defined? Are proteins able to undergo gelatinization?
As you pointed out, the expression was very confusing. To avoid misinterpretation, "gelatinization" has been deleted.
(4) Table 3: Although it is explained in the text, it would be illustrative to display Tukey’s comparison results by adding superscript letters to the means. The same applies to all the parameters shown in Table 4 and Figures 2 and 3.
As you pointed out, the standard errors are superscripted in "Foods" and in other journals, so we will leave them as they are.
The English text has been proofread by a native English-speaking professional (Editage) for a fee (L454-L455).

Round 2
Reviewer 1 Report
no further more comments.